

# Are some individuals generally more behaviorally plastic than others? An experiment with sailfin mollies

Julie Gibelli[1], Nadia Aubin-Horth[2] and Frédérique Dubois[1]

[1] Département de Sciences Biologiques, Université de Montréal, Montréal, QC, Canada
[2] Département de Biologie et Institut de Biologie Intégrative et des Systèmes (IBIS), Université Laval, Québec, QC, Canada

## ABSTRACT

Individuals within the same population generally differ among each other not only in their behavioral traits but also in their level of behavioral plasticity (i.e., in their propensity to modify their behavior in response to changing conditions). If the proximate factors underlying individual differences in behavioral plasticity were the same for any measure of plasticity, as commonly assumed, one would expect plasticity to be repeatable across behaviors and contexts. However, this assumption remains largely untested. Here, we conducted an experiment with sailfin mollies (*Poecilia latipinna*) whose behavioral plasticity was estimated both as the change in their personality traits or mating behavior across a social gradient and using their performance on a reversal-learning task. We found that the correlations between pairwise measures of plasticity were weak and non-significant, thus indicating that the most plastic individuals were not the same in all the tests. This finding might arise because either individuals adjust the magnitude of their behavioral responses depending on the benefits of plasticity, and/or individuals expressing high behavioral plasticity in one context are limited by neural and/or physiological constraints in the amount of plasticity they can express in other contexts. Because the repeatability of behavioral plasticity may have important evolutionary consequences, additional studies are needed to assess the importance of trade-offs between conflicting selection pressures on the maintenance of intra-individual variation in behavioral plasticity.

## INTRODUCTION

Over the past decades, considerable interest has been devoted to the study of animal personality, which is *defined* as a suite of individual differences in behavior that are consistent over time and contexts (*Sih & Bell, 2008*). Numerous studies on animal personality, notably, have demonstrated that individuals within the same population generally differ widely among each other not only in their behavioral traits such as their activity level, exploration tendency or aggressiveness, but also in their propensity to modify their behavior in response to changing conditions (i.e., in their level of behavioral plasticity, also referred to as contextual plasticity, activational plasticity, reversible plasticity, or responsiveness; e.g., *Stamps, 2016*). Behavioral plasticity is often considered

Corresponding author
Julie Gibelli,
julie.gibelli@umontreal.ca

advantageous (*DeWitt, Sih & Wilson, 1998*; *Gabriel et al., 2005*). Accordingly, comparative analyses have shown that bird species with relatively large brains and a high frequency of foraging innovations tend to be more successful invaders than less flexible species (*Sol & Lefebvre, 2000*; *Sol, Timmermans & Lefebvre, 2002*). Also, at the intraspecific level, two recent empirical studies found that individuals with greater behavioral plasticity had higher fitness, in terms of survival (*Toscano, 2017*) or mating success (*Montiglio et al., 2017*), compared to less plastic ones. If being plastic is advantageous, however, natural selection should erode variations in behavioral plasticity and several evolutionary explanations have thus been proposed to account for the maintenance of individual differences (*Wolf, Van Doorn & Weissing, 2008*, *2011*; *Dubois, Morand-Ferron & Giraldeau, 2010*). Although those explanations rely on different processes (e.g., state-dependent selection, frequency-dependent selection, positive feedback loops), they all implicitly assume that the proximate factors underlying individual differences in behavioral plasticity are the same for any measure of plasticity, and that an individual's plasticity, therefore, should be repeatable across behaviors and contexts (*Dingemanse & Wolf, 2013*).

In particular, in order to adjust its behavior to local conditions, an animal must detect and respond to environmental changes. Personality-related differences in behavioral plasticity could then result from individual differences in sampling behavior, sensitivity to external stimuli, risk-taking tendency or cognitive ability (*Carere & Locurto, 2011*; *Sih & Del Giudice, 2012*). Accordingly, several studies have shown that individual differences in behavioral plasticity or purported measures of plasticity such as reversal learning rate (e.g., *Guillette et al., 2010*; *Lucon-Xiccato & Bisazza, 2014*; *Pintor et al., 2014*) are associated with differences in personality traits (*Gibelli & Dubois, 2017*; *Guayasamin, Couzin & Miller, 2017*; *Guido et al., 2017*; *Mazza et al., 2018*; see also reviews by *Mathot et al., 2012* and *Stamps, 2016*), hence suggesting that certain behavioral traits or particular skills that are associated with them could prevent individuals from exhiting optimal behavioral plasticity. By contrast, however, several authors reported the opposite effect or no effect of personality type on how individuals respond to changing stimuli (e.g., *Logan, 2016b*; *Bensky et al., 2017*), whereas the few studies that have measured behavioral plasticity for the same individuals in several traits or contexts (*Biro, Beckmann & Stamps, 2010*; *Morand-Ferron, Varennes & Giraldeau, 2011*; *Logan, 2016a*) or from different proxies (*Brucks et al., 2017*; *Johnson-Ulrich, Johnson-Ulrich & Holekamp, 2018*) found weak or no support for a general plasticity. For instance, despite most individuals becoming more active and bold as temperature increased in damselfish (*Pomacentrus bankanensis*), *Biro, Beckmann & Stamps (2010)* found that the degree of plasticity in activity of a given individual was unrelated to its level of plasticity in boldness. In the same vein, *Morand-Ferron, Varennes & Giraldeau (2011)* found that the most plastic individuals were not the same in two foraging games in nutmeg mannikins (*Lonchura punctulata*) while *Mitchell & Biro (2017)* reported that zebrafish (*Danio rerio*) that displayed high responsiveness to temperature were not more responsive to food deprivation. A lack of correlation in individual plasticity across contexts or behaviors could be explained by at least the two following causes.

First, behavioral plasticity would not be a general feature of an individual because its ability to perceive changes in environmental conditions and adjust its behavior accordingly, is affected by the magnitude of the changes. More precisely, the speed–accuracy trade-off hypothesis (*Chittka, Skorupski & Raine, 2009*) predicts that fast-exploring individuals should make faster but less accurate decisions compared to slow-exploring individuals that invest more time in decision process. Such a trade-off between speed and accuracy, however, would exist only when the variations in environmental conditions are subtle and hence require considerable sampling effort to be detected. Conversely, when changes in external stimuli are easy to detect, fast-exploring individuals might be capable of making fast and accurate decisions (*Mamuneas et al., 2015*). According to this idea, an individual that explores its environment superficially would then be unlikely to adjust its behavior to local conditions when the changes in environmental conditions are subtle but could exhibit greater plasticity when the magnitude of the change is large. Second, low individual consistency in plasticity across behaviors could result from physiological constraints that would reduce the capacity of individuals to adjust hormonally-mediated behaviors to changing environmental conditions (*Hau & Goymann, 2015*). For instance, experimentally elevated levels of testosterone have been shown to increase courtship behavior and aggression but to reduce parental care in several species (*McGlothlin, Jawor & Ketterson, 2007*), hence suggesting that its physiological state could, in some cases, prevent an individual to respond optimally to changing environmental conditions. Whether the level of behavioral plasticity expressed by an individual is limited by its ability to detect environmental changes or by its physiological state, we would then expect individual differences in behavioral plasticity to be associated with individual differences in personality traits. However, the strength and direction of the correlations should vary depending on the measure of plasticity and/or on the magnitude of the change in environmental conditions.

In order to test whether some individuals are generally more behaviorally plastic than others, as commonly assumed, or if their ability to adjust their behavior in response to changing conditions is trait and/or context-dependent, we conducted an experiment with sailfin mollies (*Poecilia latipinna*). For each fish, we measured its behavioral plasticity both as the change in its personality traits or mating behavior in response to changes in environmental stimuli and as its performance on a reversal-learning task. Then, we tested whether the different measures of behavioral plasticity were correlated between each other.

## MATERIALS AND METHODS

### Animals and housing conditions

We used 57 (41 males and 16 females) sexually mature sailfin mollies aged 4 months that had not participated in any previous experiment: 33 males were used as focal individuals whereas eight other males and 16 females served as audience individuals. All individuals came from a commercial fish supplier (Mirdo Importations Canada Inc., Montreal, QC, Canada). The fish were kept in brackish water at 24 ± 0.5 °C with a 12:12 photoperiod and were fed with spirulina flakes (2% of their weight) and brine

shrimps twice a day. Outside the experiments, focal males were housed either individually (15 individuals) in three L tanks (10 × 20 × 15 cm) or socially at a density of 3 ± 1 fish (18 individuals) in six L tanks (20 × 20 × 15 cm), whereas the other fish that served as audience individuals were housed in six L tanks with a maximum of 4 ± 1 same-sex congeners, for males and females, respectively. The experiments were conducted in June 2015 at the LARSEM (Laboratoire aquatique de recherche en sciences environnementales et médicales, Université Laval, Québec, QC, Canada), were approved by the Animal Care Committee of the Université Laval (animal care permit #2015027-1) and conformed to all guidelines of the Canadian Council on Animal Care.

## Male body length

The week before the beginning of the experiments, all male sailfin mollies were tranquilized with a low dose (five mg/L) of anaesthetic (TMS-222), tagged with an elastomer tag (Northwest Marine Technologies, Shaw Island, WA, USA) and photographed. Body length (distance from the tip of the snout to the base of the tail fin) was then measured (precision 0.1 mm) using Adobe Photoshop CC. The measurements were taken three times on each individual by the same examiner and then averaged. Based on the distribution of male body length, we then defined three categories of males: small (<5.90 cm), medium (5.90–6.56 cm), and large (>6.56 cm).

## Measures of plasticity

All focal males were then tested in three experiments (i.e., mating behavior, personality, and reversal learning) in the same order. The experiments were performed between 8:30 am and 3:30 pm. All trials were conducted in a 54 L tank (60 l × 30 w × 30 h/cm) whose sides were covered by black cloth in order to minimize disturbance and were recorded with a digital camera (JVC model GZ-MS120).

### Mating behavior

As in *Fraser et al. (2014)*, we quantified the mating behavior of each focal male first in a control condition (i.e., with three females) and then in a competitive condition (i.e., in presence of three male competitors and six females), with a 1-week delay between the two conditions. For the competitive condition, we insured that the three males were of different sizes (i.e., one small, one medium, and one large fish) and that the size difference between two males was larger than 0.6 cm. Before being tested, the focal fish were isolated from their conspecifics for 24 h. For each test, we successively introduced into the tank the audience females, the audience males, when applicable, and finally the focal male. Then, during a 10-min observation period, we scored the male's sexual behavior as (1) the number of nibbles in the vicinity of a female's urogenital opening and (2) the number of gonoporal thrusts, as both traits have been shown to be plastic in response to changes in competitive conditions (*Travis & Woodward, 1989*; *Fraser et al., 2014*). After the tests were completed, we calculated the plasticity in mating behavior of each male as the difference in the number of nibbling bites or gonoporal thrusts between the control and competitive conditions both in absolute and relative values. For subsequent analysis, however, we used only plasticity in the number of nibbling bites as only this

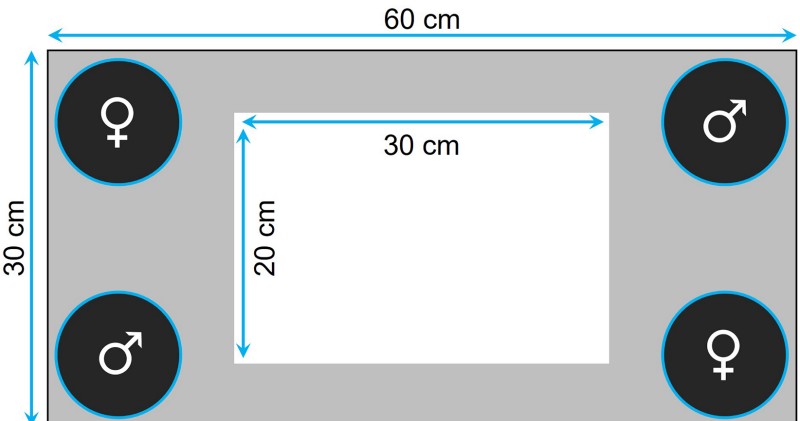

**Figure 1 Top view of the tank used to measure exploration and anxiety.** The four containers, placed in each corner of the tank, housed a single individual when the fish was tested with an audience or were kept empty otherwise. The gray area delimits the area considered to measure anxiety in the alone treatment.

trait was significantly affected by the condition though both traits were correlated (Supplementary Material S1). Given that the males were housed with a number of conspecifics that varied between zero and four before being isolated for 24 h and then tested, we insured that the housing condition of the fish had no effect on their mating behaviors (Supplementary Material S1).

## Personality

We conducted two behavioral tests that are commonly used to assess personality traits (*Réale et al., 2007*). Specifically, the open-field test was used to measure exploration and thigmotaxis, an indicator of anxiety (*Maximino et al., 2010*), while the novel object test was used to measure neophobia. All individuals performed the two tests in the same order (i.e., open-field test and novel object test) and each test was replicated twice without an audience (T1, T2) and twice with a randomly selected audience (T3, T4). For each test, the interval between the alone and audience treatments was 7 days, while the delay between two replicates for a given treatment varied between 6 and 24 h.

### Open-field test: exploration and anxiety

Prior to testing, the fish were familiarized with the tank for 30 min twice a day for 7 consecutive days. The tank was equipped with four transparent plastic containers ballasted with four to five sterilized black rocks, one in each corner, where we could confine the audience individuals (Fig. 1). Specifically, for the audience treatment, four individuals (i.e., two males and two females) were placed in the containers (i.e., one individual per container) before the focal fish was introduced, while the containers were kept empty for the alone treatment. The containers were closed at the top with a net, thereby preventing physical, but not chemical, interactions. At the beginning of a trial, we first introduced the focal fish in the center of the tank and then, once we had removed the landing net from the top of the tank, we monitored its movements for 5 min using a custom software. The images were divided into squares of $35 \times 35$ pixels

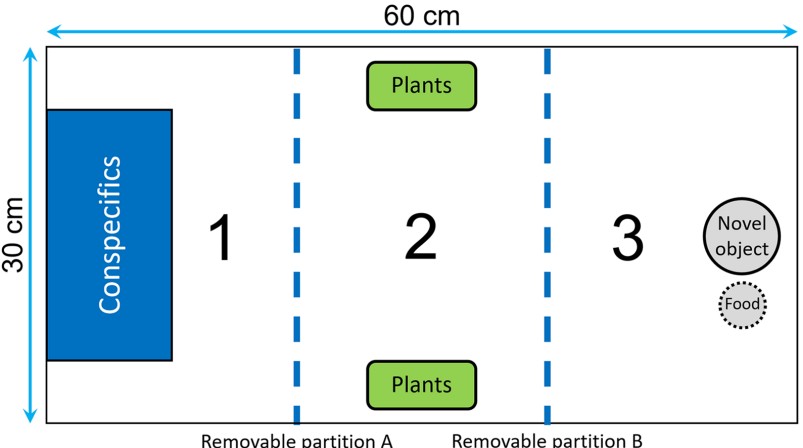

**Figure 2  Top view of the tank used to measure neophobia.** The tank was divided into three identical sections (30 × 20 cm) by two opaque removable partitions (blue dotted lines). Section 1 contained a transparent box used to confine the audience males, Section 2 was equipped with two plastic plants that could be used as a shelter, and Section 3 contained a novel object and a food box.

(around 5 × 5 cm), so that we could estimate, for each focal fish, its exploration tendency, as the number of different squares visited during the 5-min observation period.

The same recorded data were also used to measure thigmotaxis as the total number of squares visited by the fish that were located at the periphery of the walls and the containers (i.e., gray area in Fig. 1) when tested alone.

*Novel object test: neophobia*

We divided the test tank into three sections by two opaque removable partitions that could be moved with pulley system in order to avoid any disturbance (Fig. 2): the first section contained a transparent rectangular container (20 × 10 × 15 cm) where we could confine the audience individuals; the central section contained two plastic plants that could be used as a shelter; the end section contained a novel object that consisted of a small colored figurine (around four cm in diameter) placed one cm from the food box. The food box, which was made of a small net and contained brine shrimps, was used as an attractive olfactory signal to encourage the fish to approach the novel object. At the beginning of a trial, we introduced the focal fish into the first section of the tank, and then we gently removed the partitions A and B after 5 and 20 min, respectively (Fig. 2). We let the fish acclimatize for 20 min before removing partition B to ensure that no fish would swim erratically and enter in contact with the novel object by chance. Then, we estimated neophobia as the latency to come within one cm of the novel object. A trial ended when the focal fish had approached the food or after 20 min, whichever occurred first. For the audience treatment, two males were confined in the container before the focal fish was introduced, while the container was kept empty for the alone treatment. We placed the audience in the first section of the aquarium to maximize the chance of observing differences between the two treatments in the case where some fish would exhibit maximum latencies when tested alone.

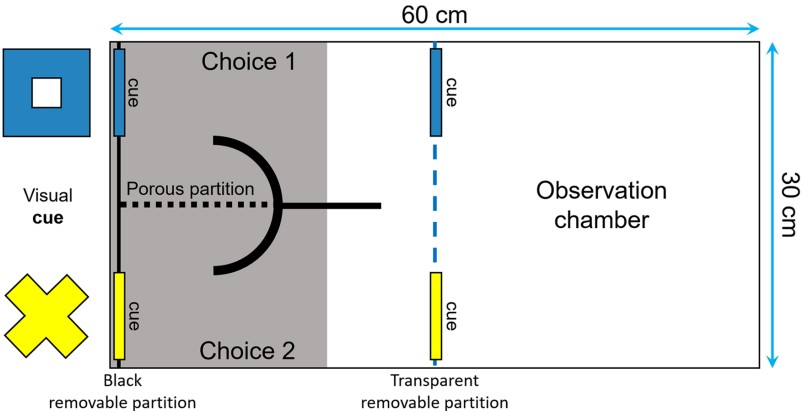

**Figure 3 Top view of the tank used to measure reversal learning speed.** The tank was divided in two sections (30 × 30 cm) by a transparent removable partition (blue dotted line): an observation compartment and a choice compartment divided into two corridors. Two visual cues (a blue square and a yellow cross of around 4 × 4 cm) were placed both in front and at the end of each corridor to ensure that the fish could see them from the observation compartment and learn which cue was rewarded.

After the two tests were completed, we estimated for each fish its levels of plasticity under three different social gradients as the difference in absolute and relative values between (i) the two replicates of the alone treatment (e.g., |T1 − T2|) (gradient A), (ii) the two replicates of the audience treatments (e.g., |T3 − T4|) (gradient B), and (iii) the mean trait value in the alone treatment and the mean trait value in the audience treatment (e.g., |(T1 + T2 − T3 + T4)/2|) (gradient C). For each individual, therefore, we had a maximum of seven absolute or relative measures of plasticity in personality traits (i.e., one for anxiety, three for exploration and three for neophobia). For statistical analysis, however, differences between two measures with the highest possible value ($n = 12$) were replaced by missing data in order to avoid ceiling effects.

### Reversal learning

The fish were trained to perform an associative learning task and then were tested for reversal learning, often viewed as a measure of behavioral flexibility by psychologists and neuroscientists (*Fellows & Farah, 2003*; *Izquierdo et al., 2007*; *Haluk & Floresco, 2009*; *Izquierdo & Jentsch, 2012*). The tests were performed in a tank divided with a transparent removable partition into one observation compartment and one choice compartment that had two corridors, separated from each other by a porous partition (Fig. 3). Two visual cues of different shapes and colors (i.e., a yellow cross and a blue square of around 4 × 4 cm) were placed at the end of each corridor. Two identical symbols were also placed in front of the corridors, to ensure that the fish could see them from the observation compartment. Prior to the start of the learning tests, following 2 days of food deprivation, the fish were habituated to the apparatus during three periods of 30 min each. We insured that they were all able to explore both corridors of the choice compartment to get food.

The fish were tested after 24 h of food deprivation (unless they made no choice in the first 20 min in which case they were deprived of food for another 24-h period) until they

choose the correct corridor six consecutive times. During the test, the position of each cue remained constant from trial to trial, but the rewarded cue, as well as its position, were counterbalanced across subjects. At the beginning of each trial, the focal fish was placed in the observation compartment for 5 min or until it has ceased freezing or swimming erratically. We then placed three spirulina flakes on the side of the rewarded cue and gently lifted the removable partition to allow the fish to enter the choice compartment and make a decision. To prevent the food from scattering in the aquarium and from being visible from the observation compartment, it was placed behind a curved PVC partition. We considered that the fish had made a choice once half of its body had entered the choice compartment. If the fish succeeded, it could eat the three spirulina flakes before returning to the observation chamber. Otherwise, the observer gently activated the removable partition to encourage the fish to return into the observation compartment. One week later, the fish were tested for reversal learning after 2 days of food deprivation. To ensure that the fish remembered which cue was rewarded in the acquisition phase, we retested them using the same procedure as above until they choose the correct corridor six consecutive times. All individuals were then tested for reversal learning 2–2.5 h after they reached this criterion, except two fish that refused to participate in the reversal learning trials and hence were tested again after 24 h. *The reversal learning* trials were performed in the same manner as described above, except that the previously unrewarded cue was now rewarded, and the previously *rewarded cue was* unrewarded. Each fish was tested until it chose the correct corridor six consecutive times. The number of trials required by each fish to reach this criterion was then used as a proxy of its plasticity.

## Statistical analyses

Our complete data set contained five missing values. In three occasions, the missing values were due to technical problems. In the two other cases, we replaced the anxiety score of a fish by a missing value because it stayed immobile in the middle of the test arena for the entire duration of the test, while another individual was never able to explore the choice compartments of the learning apparatus during the familiarization period and hence had no value for the reversal learning test.

To explore how the nine measures of plasticity were related to each other, we used Spearman rank correlations to test for pairwise correlations and applied the improved Bonferroni correction to account for multiple comparisons (*Hochberg, 1988*). Correlations were calculated using both absolute and relative measures of plasticity. We also performed a principal component analysis (PCA) on the nine measures of plasticity in absolute value to test for a general plasticity that would be explained by a single factor. Finally, we tested whether the level of plasticity in exploration and neophobia differed among the social gradients for a given individual with paired-*t* tests and explored whether the amount of plasticity in personality displayed in response to gradients A and B depended on the level of expression of the trait during the first exposure to the stimuli (i.e., T1 or T3) by using Spearman rank correlations.

All statistical analyses were conducted on R 3.2.4 (*R Core Team, 2016*). PCA was performed on the standardized data and using the dudi.pca function.

**Table 1 Pairwise Spearman rank correlations among the nine measures of behavioral plasticity estimated as the differences in absolute value.**

| | | Neophobia A | Exploration A | Anxiety A | Neophobia B | Exploration B | Neophobia C | Exploration C | Mating behavior |
|---|---|---|---|---|---|---|---|---|---|
| Exploration A | $r_s$ | 0.369 | | | | | | | |
| | $p$ | 0.053 | | | | | | | |
| | $n$ | 28 | | | | | | | |
| Anxiety A | $r_s$ | 0.530 | 0.281 | | | | | | |
| | $p$ | 0.004 | 0.148 | | | | | | |
| | $n$ | 27 | 28 | | | | | | |
| Neophobia B | $r_s$ | 0.259 | 0.309 | −0.217 | | | | | |
| | $p$ | 0.221 | 0.142 | 0.319 | | | | | |
| | $n$ | 24 | 24 | 23 | | | | | |
| Exploration B | $r_s$ | 0.171 | 0.277 | −0.115 | 0.429 | | | | |
| | $p$ | 0.366 | 0.132 | 0.560 | 0.029 | | | | |
| | $n$ | 30 | 31 | 28 | 26 | | | | |
| Neophobia C | $r_s$ | 0.168 | 0.053 | 0.083 | 0.508 | 0.177 | | | |
| | $p$ | 0.374 | 0.780 | 0.676 | 0.008 | 0.332 | | | |
| | $n$ | 30 | 30 | 28 | 26 | 32 | | | |
| Exploration C | $r_s$ | 0.348 | 0.231 | 0.294 | 0.417 | −0.285 | 0.214 | | |
| | $p$ | 0.059 | 0.212 | 0.129 | 0.034 | 0.108 | 0.239 | | |
| | $n$ | 30 | 31 | 28 | 26 | 33 | 32 | | |
| Mating behavior | $r_s$ | 0.444 | 0.178 | 0.007 | 0.095 | 0.033 | 0.254 | 0.080 | |
| | $p$ | 0.014 | 0.339 | 0.974 | 0.645 | 0.856 | 0.160 | 0.657 | |
| | $n$ | 30 | 31 | 28 | 26 | 33 | 32 | 33 | |
| Reversal learning | $r_s$ | 0.019 | 0.305 | 0.386 | 0.089 | 0.041 | −0.047 | 0.154 | −0.285 |
| | $p$ | 0.922 | 0.101 | 0.043 | 0.667 | 0.824 | 0.798 | 0.401 | 0.113 |
| | $n$ | 30 | 30 | 28 | 26 | 32 | 32 | 32 | 32 |

Note:
Reversal learning scores (i.e., the number of trials to reach the learning criterion in the reversal task) were multiplied by −1 in such a way that the subjects that were faster to reverse had the highest scores.

## RESULTS

Although individual measures of plasticity in absolute value were positively correlated in the majority (31/36) of pairwise comparisons (Table 1), none of these correlations were significant (Bonferroni corrected $\alpha = 0.001$). Explicitly, the correlations were at best moderate (i.e., between 0.4 and 0.59) with an average correlation equal to 0.174. The correlations were even smaller when considering the differences in relative value (Table 2) as always at least 21% (and up to 50%) of the focal fish adjusted their behavior to the changes in social conditions in the opposite direction to that of their congeners. For instance, 21 fish were more exploratory when tested with an audience than alone, but 12 other fish on the contrary had a higher exploration score in the alone treatment. Thus, our findings indicate that an individual's level of plasticity was weakly repeatable across contexts and behaviors. This conclusion is further supported by the fact that four principal *components* with eigenvalues greater than one were extracted from the PCA, with each of them explaining only a small portion of the total variance (Table 3).
**Table 2 Pairwise Spearman rank correlations among the nine measures of behavioral plasticity estimated as the differences in relative value.**

| | | Neophobia A | Exploration A | Anxiety A | Neophobia B | Exploration B | Neophobia C | Exploration C | Mating behavior |
|---|---|---|---|---|---|---|---|---|---|
| Exploration A | $r_s$ | −0.221 | | | | | | | |
| | $p$ | 0.258 | | | | | | | |
| | $n$ | 28 | | | | | | | |
| Anxiety A | $r_s$ | −0.005 | −0.098 | | | | | | |
| | $p$ | 0.981 | 0.620 | | | | | | |
| | $n$ | 27 | 28 | | | | | | |
| Neophobia B | $r_s$ | 0.003 | −0.391 | 0.188 | | | | | |
| | $p$ | 0.987 | 0.059 | 0.391 | | | | | |
| | $n$ | 24 | 24 | 23 | | | | | |
| Exploration B | $r_s$ | −0.165 | 0.174 | −0.328 | −0.189 | | | | |
| | $p$ | 0.384 | 0.350 | 0.089 | 0.355 | | | | |
| | $n$ | 30 | 31 | 28 | 26 | | | | |
| Neophobia C | $r_s$ | 0.000 | 0.260 | −0.140 | −0.018 | 0.312 | | | |
| | $p$ | 0.999 | 0.166 | 0.479 | 0.930 | 0.082 | | | |
| | $n$ | 30 | 30 | 28 | 26 | 32 | | | |
| Exploration C | $r_s$ | 0.189 | −0.106 | −0.269 | 0.166 | −0.160 | −0.02 | | |
| | $p$ | 0.318 | 0.572 | 0.166 | 0.416 | 0.372 | 0.914 | | |
| | $n$ | 30 | 31 | 28 | 26 | 33 | 32 | | |
| Mating behavior | $r_s$ | −0.275 | −0.167 | −0.035 | 0.115 | 0.114 | 0.312 | −0.095 | |
| | $p$ | 0.141 | 0.368 | 0.858 | 0.576 | 0.527 | 0.082 | 0.598 | |
| | $n$ | 30 | 31 | 28 | 26 | 33 | 32 | 33 | |
| Reversal learning | $r_s$ | 0.397 | −0.103 | −0.032 | −0.275 | −0.099 | 0.088 | −0.325 | 0.328 |
| | $p$ | 0.030 | 0.588 | 0.870 | 0.174 | 0.591 | 0.631 | 0.069 | 0.067 |
| | $n$ | 30 | 30 | 28 | 26 | 32 | 32 | 32 | 32 |

Note:
Reversal learning scores (i.e., the number of trials to reach the learning criterion in the reversal task) were multiplied by −1 in such a way that the subjects that were faster to reverse had the highest scores.

Furthermore, the four first principal components had positive and negative loading values (Table 3), which reflects the fact that the different measures of plasticity were not equivalent.

There were differences in the level of plasticity expressed by individuals depending on the social gradient they were exposed to. Most notably, the magnitude of the changes in behavioral responses was smaller when the fish were tested twice without an audience compared to when they were observed two times by different audience individuals or when we compared the average response with and without an audience (Fig. 4). However, only the difference between gradient A and C was significant for measures of neophobia ($t_{29} = −2.347$, $p = 0.026$). Finally, for the three personality traits measured, we found a significant association between the amount of plasticity exhibited and the value of the trait during the first trial (Neophobia, A: $r_s = 0.644$, $n = 30$, $p < 0.001$; B: $r_s = 0.526$, $n = 26$, $p = 0.006$; Exploration, A: $r_s = 0.561$, $n = 31$, $p = 0.001$; B: $r_s = 0.488$, $n = 33$, $p = 0.004$ & Anxiety, A: $r_s = 0.481$, $n = 28$, $p = 0.010$;

**Table 3 Loadings on the four extracted factors.**

| Plasticity measure | PC1 | PC2 | PC3 | PC4 |
|---|---|---|---|---|
| Neophobia A | 0.705 | 0.502 | −0.298 | 0.043 |
| Exploration A | 0.710 | 0.337 | −0.035 | −0.251 |
| Anxiety A | 0.235 | 0.689 | 0.313 | −0.005 |
| Neophobia B | 0.477 | −0.546 | 0.551 | 0.122 |
| Exploration B | 0.498 | −0.021 | −0.026 | 0.807 |
| Neophobia C | 0.514 | −0.455 | 0.110 | 0.129 |
| Exploration C | 0.606 | −0.241 | 0.452 | −0.467 |
| Mating behavior | 0.568 | −0.129 | −0.704 | −0.157 |
| Reversal learning | −0.043 | 0.716 | 0.439 | 0.105 |
| Eigenvalues | 2.487 | 1.934 | 1.397 | 1.002 |
| % of variance | 27.6% | 21.5% | 15.5% | 11.1% |

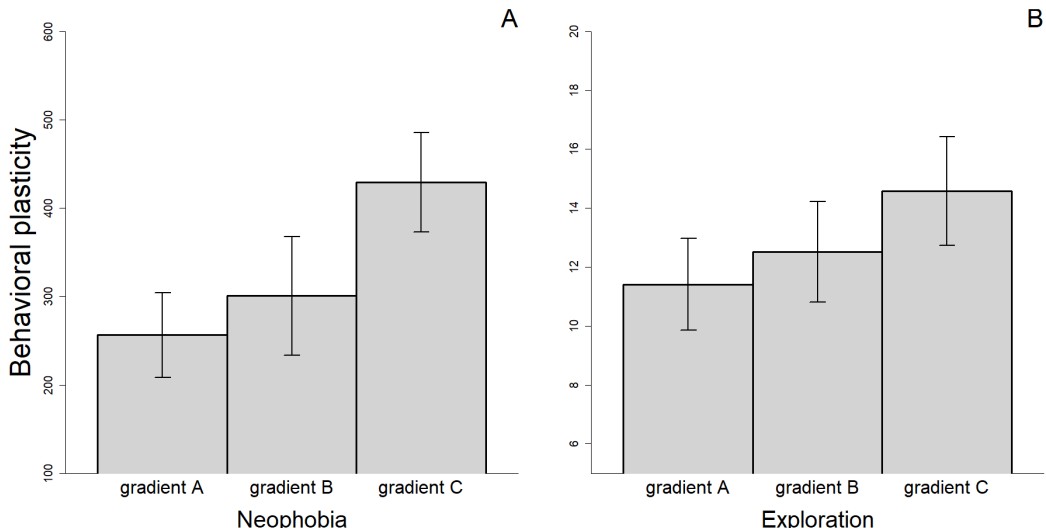

**Figure 4 Mean (±SEM) behavioral plasticity in neophobia (A) and exploration (B) measured under three different social gradients.** Behavioral plasticity was estimated as the difference in absolute value between (i) the two replicates of the alone treatment (gradient A), (ii) the two replicates of the audience treatment (gradient B), and (iii) the mean trait value in the alone treatment and the mean trait value in the audience treatment (gradient C).

see also Supplementary Material S2). Individuals that exhibited the greatest plasticity, therefore, showed greater expression of the trait when they were exposed to the stimuli for the first time.

## DISCUSSION

Using sailfin mollies, we tested whether some individuals are generally more behaviorally plastic than others, or if an individual's ability to adjust its behavior in response to changing conditions is trait and/or context-dependent. We found that the correlations between pairwise measures of plasticity were non-significant, thus indicating that the most plastic individuals were not the same in all the tests. Although our sample size is small, and the

power to detect significant associations is relatively low, this conclusion is supported by the fact that the correlations were only weak or moderate in strength. The correlations were particularly weak when considering not only the magnitude but also the direction of behavioral changes, as individuals differed widely from each other in how they were affected by the treatments. Two non-exclusive explanations could account for our finding.

First, the lack of correlation between the different measures of plasticity could result from differences among individuals in their internal state that would have affected their willingness to express behavioral plasticity in response to environmental conditions (*Wolf & Weissing, 2010*). Indeed, the extent to which an animal actually changes its behavior in response to changing stimuli is a measure of realized plasticity and hence does not necessarily reflect potential plasticity (*Stamps & Krishnan, 2014*; *Stamps, 2016*). Individuals with high potential plasticity, therefore, might have exhibited under certain conditions or for certain traits little plasticity, resulting in a weak association between the different measures of realized plasticity. Our results partly support this hypothesis, as they suggest that individuals would adjust the magnitude of their behavioral responses depending on the benefits of plasticity. Indeed, we found that the fish exhibiting the greatest plasticity in relative value in response to the social gradients A and B showed greater expression of the trait (i.e., were the most neophobic, exploratory, and anxious individuals) when they were exposed to the stimuli for the first time, probably because those individuals tend to overestimate the danger when confronted to a novel situation, thereby reacting inappropriately. In accordance to this idea, *Mitchell & Biro (2017)* also found that active individuals that deplete energy reserves at a faster rate than less active fish, were more responsive to food deprivation. Nonetheless, individuals that were the most plastic in their activity level in response to changes in food abundance were not the most plastic in response to changes in temperature (*Mitchell & Biro, 2017*). The importance of past behavior on the expression of behavioral plasticity, therefore, would depend on the environmental gradient used and/or on the trait being measured. Given the fact that all subjects in our study experienced the trials in the same order, however, this explanation alone cannot account for the absence of significant correlations among the different measures of plasticity.

Thus, another non-mutually exclusive interpretation of our findings is that behavioral plasticity is not a general characteristic of individuals, such that individuals expressing high behavioral plasticity in one context could be limited in the amount of plasticity they can express in other contexts. Particularly, different measures of behavioral plasticity taken under different conditions or using different traits, would not be equivalent because they are controlled by different neural, physiological or behavioral mechanisms, so that some behavioral traits might be more constrained than others. Supporting this idea, the PCA analysis revealed that the first component accounted for less than 28% of the total variance, which is low compared to results that are usually interpreted as support for a general factor (see *Shaw & Schmelz, 2017* and references therein). Our interpretation is also supported by the fact that the rate of reversal learning, which is regularly used as a proxy for behavioral plasticity (e.g., *Guillette et al., 2010*; *Lucon-Xiccato & Bisazza, 2014*; *Pintor et al., 2014*), was very weakly or not correlated with the other measures of individual

plasticity. There is evidence that reversal-learning tasks assess distinct neurological abilities from other tasks used to measure behavioral plasticity, such as set-shifting or self-control tasks (*Audet & Lefebvre, 2017*). This might explain why different measures of plasticity are uncorrelated. Accordingly, we found that reversal learning performance loaded negatively on the first principal component contrary to the eight other measures of plasticity that had positive loadings. Individuals capable of inhibiting a previously rewarded behavior more quickly, therefore, would possess particular skills that would not be associated with greater plasticity in personality. In particular, our results are consistent with the hypothesis that plasticity in personality traits may be restrained by the inability of individuals to detect environmental changes. Indeed, as anticipated, fish exhibited on average greater plasticity between the alone and audience treatments than between the two trials of the audience treatment for both personality traits measured (e.g., exploration and neophobia). Furthermore, the correlation between the levels of plasticity in exploration and neophobia was stronger when plasticity was measured as the difference between the two trials with a randomly selected audience than between the audience and alone treatments. This finding indicates that individuals were more consistent in their plasticity when the changes in environmental conditions were subtle, as only more neophobic (and less exploratory) individuals that explore their environment slowly and accurately could then detect and respond to such environmental changes (*Sih & Del Giudice, 2012*). Thus, our results suggest that trade-offs between conflicting selection pressures might maintain intra-individual variation in behavioral plasticity. However, in order to confirm this hypothesis, further studies will be required to demonstrate the adaptive nature of intra-individual differences in the expression of behavioral plasticity. Such studies will consist notably in (i) testing the effect of spatial and/or temporal environmental heterogeneity on the repeatability of behavioral plasticity, and (ii) measuring the fitness consequences of individual plasticity in various contexts.

## CONCLUSIONS

In conclusion, although we cannot conclude from this study to what extent individuals are limited in their level of behavioral plasticity by neural and/or physiological constraints, our findings contradict the widespread (though largely untested) idea that some individuals would be generally more behaviorally plastic than others (*Dingemanse & Wolf, 2013*, but see *Audet & Lefebvre, 2017*). Whether behavioral plasticity is a repeatable trait at the individual level may have important evolutionary consequences, for instance through affecting population stability or persistence (*Wolf & Weissing, 2012*; *Sih et al., 2012*). Additional studies, therefore, should be conducted to investigate the importance of trade-offs between conflicting selection pressures on the maintenance of intra-individual variation in behavioral plasticity and examine the fitness consequences of behavioral plasticity in different contexts.

## ACKNOWLEDGEMENTS

We would like to thank Richard Khoury who designed a custom software, allowing us to conduct more precise observations on the video recordings of sailfin mollies.

### Funding

This work was supported by research grants awarded to Frédérique Dubois and to Nadia Aubin-Horth (Discovery Grants Program) from the Natural Sciences and Engineering Research Council of Canada. Julie Gibelli received financial support through a scholarship from the Principauté de Monaco. The funders had no role in study design, data collection and analysis, decision to publish, or preparation of the manuscript.

### Grant Disclosures

The following grant information was disclosed by the authors:
Frédérique Dubois and to Nadia Aubin-Horth (Discovery Grants Program) from the Natural Sciences and Engineering Research Council of Canada.
Principauté de Monaco.

### Competing Interests

The authors declare that they have no competing interests.

### Author Contributions

- Julie Gibelli conceived and designed the experiments, performed the experiments, analyzed the data, prepared figures and/or tables, authored or reviewed drafts of the paper, approved the final draft.
- Nadia Aubin-Horth conceived and designed the experiments, contributed reagents/materials/analysis tools, approved the final draft.
- Frédérique Dubois conceived and designed the experiments, analyzed the data, contributed reagents/materials/analysis tools, authored or reviewed drafts of the paper, approved the final draft.

### Animal Ethics

The following information was supplied relating to ethical approvals (i.e., approving body and any reference numbers):

The experiments were conducted in June 2015 at the LARSEM (Laboratoire aquatique de recherche en sciences environnementales et médicales, Université Laval, Québec, QC, Canada), and were approved by the Animal Care Committee of the Université Laval (animal care permit #2015027-1) and conformed to all guidelines of the Canadian Council on Animal Care.

### Data Availability

Gibelli, Julie; Aubin-Horth, Nadia; Dubois, Frédérique (2018): Are some individuals generally more behaviorally plastic than others? An experiment with sailfin mollies. figshare. Dataset. https://doi.org/10.6084/m9.figshare.6237518.v2.

## Supplemental Information

Supplemental information for this article can be found online at http://dx.doi.org/10.7717/peerj.5454#supplemental-information.

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
