# Peer review of "Are some individuals generally more behaviorally plastic than others? An experiment with sailfin mollies"

_PeerJ, doi:10.7717/peerj.5454_

## Round 0.1 · original submission · Major Revisions

I agree with the reviewers that you have tackled an important topic. You have extended this research to a new species, the studies are well designed, and your article is well-written. However, both of the reviewers express some concern with your statistical approach and your definition of personality. The reviewers have provided very thorough and helpful comments, along with some recommendations for the inclusion of previous research so I will not reiterate their advice. I would like to invite you to revise the manuscript, being careful to address the reviewers’ concerns to the best of your ability. In particular, you will need to address the notion of circularity in treating ‘personality’ as measured by neophobia and exploration with flexibility. I like the idea of contrasting flexibility in the face of different environmental contexts with flexibility between types of behavior and over time, but I do agree that some of the complexities of studying flexibility using different measures are lost in your discussion.

I have a few minor comments of my own:

If the number of nibbles differ between control and competitive conditions, could this be because of different access or distraction from rivals in competitive conditions and not flexibility per se?

Were there differences between males housed singly and group-housed especially with regard to mating behavior?

Do not use “since” or “while” unless in temporal context.
In addition to the references suggested by the reviewers, I am surprised you do not discuss some of the work by Daniel Sol.

This paper might also be useful:

Watters, J. V., & Meehan, C. L. (2007). Different strokes: Can managing behavioral types increase post-release success? Applied Animal Behaviour Science, 102(3-4), 364-379. doi:http://dx.doi.org/10.1016/j.applanim.2006.05.036

Reviewer 1 ·

Basic reporting

This is a nice little study of relationships, across individuals, between different types of behavioral plasticity. The study focuses on contextual plasticity: the behavior expressed by the same individuals when they are tested in two different social contexts. It also includes an assay of a different type of plasticity: a measure of learning rate in a reversal learning task, estimated using a standard method in the field (number of trials to criteria).
At present there are a lot of ideas floating around about reasons why one might expect particular types of plasticity to be correlated with one another across individuals. However, data on this topic is still sparse. So the current study should provide some much-needed information on this topic.

See comments below on suggestions on references that might be added to the paper, and suggestions on ways to reorganize the introduction

Experimental design

The experimental design used to measure the behavior of the subjects seems fine, with evidence that the authors have taken a number of steps to control for spurious internal and external factors that might have affected the behavior expressed by their subjects.
My main concerns have to do with the statistical analyses. It seems a shame that the authors were not able to use a multivariate analysis which would allow them to simultaneously analyze the variation within contexts (e.g. variation across the 2 alone treatments) and the variation across contexts (e.g. variation in the behavior expressed in the different social treatments). At present, the large number of comparisons necessarily make it impossible to detect statistically significant correlations of the magnitude one would expect to observe in this sort of study (see comments 10 and 13, below). Another question is whether the results of this study would have differed if plasticity had been measured based on the difference in scores, instead of being based on the absolute value of the difference in scores (see comment 6, below). The two ways of measuring plasticity will yield the same answers if the direction of plasticity is the same for all of the subjects. However it is not clear whether this is the case in the current study.

Validity of the findings

Given the number of comparisons in this study, I think that it works better as an exploratory study than as a confirmatory one. The authors present a number of results which can provide the basis for additional, more-focused, experiments on this topic. However, the current study lacks the statistical power necessary to test for correlations of the magnitude reported in previous studies of this topic. Hence, I think that it would be a mistake for the authors to try to explain why different measures of plasticity were NOT correlated with one another across individuals. Instead, they can present their results with the correlations indicated, and leave it to future research to determine whether the (mostly) positive correlations they reported do reflect underlying mechanisms which affect different types of behavioral plasticity. Another positive aspect of the way that they have reported their results is that other researchers who want data to test specific hypotheses about plasticity (e.g. correlations across individuals between plasticity in mating behavior and plasticity in exploratory behavior) should b able to pick that data out of their table.

Additional comments

1. Although the current article focuses on individuals, the authors may also want to look at an article that considers correlations across genotypes between different types of behavioral plasticity:
Saltz et al. 2017. Genetic Correlations among Developmental and Contextual Behavioral Plasticity in Drosophila melanogaster. Am. Nat. 190 Pages: 61-72
Some of the issues discussed in this article are relevant to the current one, e.g. the measuring correlations between contextual and developmental plasticity (see below)

2. Line 20. I am not sure why the authors consider changes in personality across a social gradient to be a 'direct' example of plasticity, and variation in learning rates to be an 'indirect' example of plasticity.

3. Lines 55-63. It might be easier for readers to follow if the hypotheses on reasons why one might expect different types of plasticity to be correlated with one another across individuals were presented first, and then studies that support or refute this hypothesis were presented. For instance, the only reason for mentioning studies that look for correlations, across individuals, between personality and plasticity is because such studies imply that the same personality trait will be related to different types of plasticity. Thus, Sih & Del Giudice 2012 (Linking behavioural syndromes and cognition: a behavioural ecology perspective. Phil. Trans. Roy Soc. B. 367, 2762–2772), is relevant because it discusses relationships across individuals between reversal learning rates, acquisition learning rates and other cognitive traits, based on the assumption that different types of cognitive traits are linked to personality traits.
For the current article, the most relevant empirical studies are not those which have looked for links between personality and plasticity, but rather those which have tested relationships across individuals between different types of plasticity. Since there currently aren't many empirical studies on THIS topic, this would be a good place to mention the ones that have been published. For instance, Mitchell&Biro 17 should be cited here, rather than waiting for the Discussion section.
At present, lines 66-67 offer a somewhat dated review of empirical studies which have looked for correlations across individuals between different types of behavioral plasticity

4. Lines 93-96. I am not sure how the results in Mathot et al. 11 support the hypothesis that physiological state prevents individuals from expressing the optimal amount of plasticity for a given trait. Although they failed to detect individual differences in the plasticity of one type of behavior (escape flights)...this was not surprising, given that they tested the subjects in flocks, and escape behavior tends to be coordinated across individuals within bird flocks.

5. Lines 106-108: " For each fish, we measured its behavioral plasticity both directly as the change in its personality traits or mating behavior in response to subtle, large or intermediate changes in environmental stimuli and indirectly as its performance on a reversal-learning task"
See 1. It is not clear why a change in a personality trait would be considered to be a 'direct' measure of behavioral plasticity whereas reversal learning would be an 'indirect' measure of behavioral plasticity.

6. Re Line 156. I get that it might be useful to measure plasticity as the absolute value of the difference between the scores expressed under two conditions, but the direction of the difference might also be relevant. For instance, if some males reduced nibbling under competitive conditions whereas other males increased nibbling under competitive conditions, then males with the same absolute plasticity scores would expressing different types of responses to the same set of conditions. So unless all of the males changed in their scores in the same direction (e.g. all of the contextual reaction norms had positive slopes as a function of level of competition) it would be a good idea to also present the results based on the actual difference in scores, and not just based on the absolute difference in scores.

7. Re Line 200. A problem re the terms 'subtle', 'medium' and 'large' is that those terms seem to be assigned based on the author's assumptions about the reasons why an individual's score might vary across treatments. For instance, 'subtle' implies that differences in external stimuli in two 'alone' treatments were responsible for differences in an individual's scores in the two treatments:
"After the 2 tests were completed, we estimated for each fish its levels of plasticity when the changes in environmental stimuli were: i) subtle (i.e. as the difference in absolute value between the 2 replicates of the alone treatment |T1-T2|)".
The problem here is that variation in the scores expressed by a given individual tested more than once in the 'same' context can vary for all sorts of reasons other than subtle variation in environmental stimuli between the tests. In particular, the individual's internal state can vary from one test to another, and this can contribute to, or be mostly responsible for, variation in its scores in this situation.
I suggest that the authors present the results without making assumptions about the extent to which the fish subjects considered the treatments to involve different stimuli.
Later (in the discussion) might be a better place to speculate about the factors that might have contributed to the plasticity in the different situations.

8. Re Line 207. It is great that the authors are aware of ceiling effects. However, just removing subjects from analysis when they had maximal possible scores for a given treatment might not fix the problem. For instance, if possible scores under low competition ranged from 0 to 1, and if the maximal observed amount of plasticity observed in any fish was +.5, then a fish with a score of .9 under low competition would be unable to express the level of plasticity expressed by another individual with a score of .5 in that condition. This is another reason why providing data on the direction of plasticity would be helpful, because a fish with a score of .9 under low competition could easily DECREASE its score by .4.
A more conservative way to avoid ceiling effects would be to determine the maximum amount of plasticity (with direction included) of plasticity expressed by any subject, and then restricting the analyses to those individuals who scores were sufficiently far from the maximum score to ensure that they could have expressed the maximum level of plasticity.

9. Re Line 219 "The fish were tested after 24h to 48h of food deprivation (depending on their satiety level) until they choose the correct corridor 6 consecutive times".
Unclear.. how was satiety level determined so that fish could be assigned to the different treatments?

10. Re Line 280. Given the very large number of comparisons, it is not surprising that none of the correlations were significant after Bonferroni correction. The authors might consider using a different test for family-wise error which might yield slightly higher p levels (e.g. Hochberg 1988, Biometrika, pp 800-802).
Also, it might make sense to present the article as an 'exploratory' study, one which might suggest profitable avenues for future research on this topic with this species. For instance, based on Table 1, it might be useful in future to look at relationships between Neophobia S and Mating Behavior, or between Reversal Learning and Anxiety S.
Conversely, it would not be reasonable to take the results presented herein as evidence of a LACK of relationships across individuals between different types of plasticity (as is currently indicated in Lines 296-298).
This is because previous studies which HAVE documented correlations across individuals between different types of behavioural plasticity usually report relatively low correlation coefficients (on the order to 0.3- 0.4). So at least some of the correlations tested in the current study might have been as strong as those reported by previous authors, but a focused experiment with greater power would be required to determine whether those correlations were significantly higher than zero.
Also, see 6, above. It is possible that correlations that were relatively weak when plasticity was measured in absolute terms would be stronger if plasticity was measured as the actual difference in scores. This is particularly true if individuals varied in the direction of plasticity.
It should be relatively easy to add correlations based on difference in scores to Table 1 by redoing the Table as a 9x9 matrix, with correlations based on difference scores above the diagonal, and correlations based on absolute scores below the diagonal.

11. Line 288. " By contrast, differences in body length were more strongly associated with differences in reversal learning speed and reproductive tactics than with differences in plasticity of personality traits: smaller fish exhibited greater plasticity in mating tactics but were slower in the reversal learning task compared with larger individuals."
Where is the statistical test that supports this finding?

12. Line 308. " Indeed, we found that individuals with the greatest plasticity estimated between the 2 trials without an audience had the longest latency". I wasn't able to find these results don't seem to be reported in the Results.

13. Line 318-320. "Since all subjects in our study experienced the trials in the same order, however, this explanation alone cannot account for the total absence of significant correlations among the different measures of plasticity".
See 10, above. In order to detect significant correlations of the same magnitude as those reported in previous studies, you would need either a much larger sample size (for the same number of comparisons) or a study with only one or two comparisons.
Currently, you have a sample size of about 30, and a p level of < .001 based on the number of comparisons. I did a quick calculation of power for this situation (power = refers to the probability that your test will find a statistically significant difference when such a difference actually exists. It is usually set at a minimum of .80).
In this study, you would need to have a correlation of at least .7 between two types of plasticity to have a reasonable chance (power = .80) of detecting a significant correlation (e.g. a correlation greater than 0). Not saying that this level of correlation across individuals between two types of behavioral plasticity is impossible, but given previous studies on this topic, it is very unlikely.

·

Basic reporting

The authors use clear, unambiguous, and professional English language throughout. The introduction and background neatly and succinctly place the context and justify the need to study behavioral plasticity in multiple contexts. However, the introduction would benefit from including reference to the term “behavioral flexibility” which refers to the same tendency to change behavior in response to changing environments. There are several papers showing that purported measures of behavioral flexibility (e.g., innovation, social learning, tool-use, see Reader, Hager, & Laland, 2011, and reversal learning/inhibitory control) do not all correlate with each other such as Logan, 2016a (reversal learning and a problem-solving task); Brucks, Marshall-Pecini, Wallis, Huber, & Range, 2017 (various measures of inhibitory control); and Johnson-Ulrich, Johnson-Ulrich, & Holekamp, 2018 (repeated innovation and inhibitory control), and probably others that I am not immediately aware of. Logan (2016b) also compares flexibility (via reversal learning) with personality traits. Generally, most studies do continue to assume that flexibility/plasticity is a single trait and studies addressing and explaining this issue are welcome.

The structure conforms to PeerJ standards, figures are relevant, high-quality, well labeled and described.
Raw data is supplied.

References:
Brucks, D., Marshall-Pescini, S., Wallis, L. J., Huber, L., & Range, F. (2017). Measures of Dogs’ Inhibitory Control Abilities Do Not Correlate across Tasks, 8(May), 1–17. https://doi.org/10.3389/fpsyg.2017.00849
Johnson-Ulrich, L., Johnson-Ulrich, Z., & Holekamp, K. (2018). Proactive behavior, but not inhibitory control, predicts repeated innovation by spotted hyenas tested with a multi-access box. Animal Cognition, 21(3), 379–392. https://doi.org/10.1007/s10071-018-1174-2
Logan, C. J. (2016a). Behavioral flexibility and problem solving in an invasive bird. PeerJ, 4, e1975. https://doi.org/10.7717/peerj.1975
Logan, C. J. (2016b). Behavioral flexibility in an invasive bird is independent of other behaviors. PeerJ, 4:e2215. https://doi.org/10.7717/peerj.2215
Reader, S. M., Hager, Y., & Laland, K. N. (2011). The evolution of primate general and cultural intelligence. Philosophical Transactions of the Royal Society of London. Series B, Biological Sciences, 366(1567), 1017–27. https://doi.org/10.1098/rstb.2010.0342

Experimental design

This paper describes original primary research within the scope of the journal.
The research question is well defined, relevant, and meaningful and the current research address an identified knowledge gap (is plasticity repeatable across contexts?).
The methods are described in sufficient detail and are generally rigorous although the measures chosen need to be justified (see line comments below).

Line 154: Please elaborate on this method to justify its inclusion as a measure of plasticity. Is it the standard way to assess sexual behavior in sailfin mollies? If so please add a citation. What do more or less nibbles indicate? Do sailfin mollies typically vary in how much they nibble, and in different contexts?

Line 158: Likewise, justify your use of exploration/anxiety and neophobia (and specific methods) to assess personality. This could simply be several representative citations and a sentence stating that these are commonly used to assess personality. (This comment applies regardless of comments in section 3).

Line 210. Please justify use of reversal learning to measure plasticity with representative citations.

Lines 200-208: “Environmental differences” are 1) uncontrollable differences between trials (please expand on what these might be) 2) differences in audience members (assuming it is a different audience each time?), and 3) alone vs audience? Please clarify this point. Are these changes that we would expect individuals should be plastic (or at least it might be advantageous to behave differently?). Please clarify 1) why these environmental conditions may or may not elicit plasticity in behavior for sailfin mollies and 2) what makes the various changes subtle, medium, and large.

Validity of the findings

Part of the data analysis has a major flaw, which hurts the discussion, but the discussion is otherwise well-stated.

The problem with the analysis is that, first, it is conceptually awkward to use measures of exploration, anxiety, and neophobia as measures of both personality (consistency over time/context) AND plasticity (change over context). Second, this creates statistical problems as you are creating correlations between two measures constructed from the same data (with the exception of personality traits, which are derived from alone trials, with the Medium-environmental change plasticity, which are derived from audience trials). In particular, correlating the personality traits with the Subtle Plasticity measures is inappropriate because the personality trait is the mean of two replicates, while the plasticity trait is the difference between the same two replicates. If this is a good measure of plasticity (e.g., subjects do vary between replicates in differing degrees), then it is NOT a valid measure of personality and using the mean is inappropriate. In order to justify using means as measures of personality traits, one must first show the data you are averaging from is repeatable. This method also means that your significant correlation between Neophobia and Neophobia-S is an artifact, not a result. If trial 1 has a very long latency, then it is unsurprising (regression to the mean) that the second trial will be much shorter. This creates subjects for which their neophobia is high (due to the high first latency skewing the mean upwards) and their plasticity is also high (also because of the high first value), and vice versa.

I would recommend choosing to use exploration, anxiety, and neophobia as either measures of personality OR plasticity, and not both. The issue of measuring personality in labs, and whether it is actually repeatable across contexts, may be a separate issue from the main topic of your paper, although you demonstrate (Fig. 4) that these “personality” traits change in different contexts, which perhaps suggests that they are not actually personality traits.
If you do want to include personality in this paper, do not use data from exploration, anxiety, and neophobia tests to measure plasticity (and instead rely on the mating behavior and reversal learning measures for plasticity. This would still yield comparable results (e.g., plasticity does not correlate in different contexts, nor does it correlate with “personality traits”) and provides ultimately the same conclusions.
Alternatively, you could continue to use exploration, anxiety, and neophobia to look at plasticity and remove Table 2 (essentially, and add body length to Table 1). This leaves you with the appreciably wide array of plasticity measures (and the environmental context), while again, drawing the same basic conclusions. (This would be my preferred solution, as it more closely aligns with your hypotheses as presented in abstract and introduction, e.g., Lines 103-110).

Other comments:
Line 273. No statistics were done on the data used to produce this figure? Looks like could be significant differences between S and L groups.
Line 331-333. This statement also support the previous ideas, so “however” is not necessary.

Additional comments

My first impression is that this is an interesting and well-written study that does a good job of analyzing plasticity in multiple contexts (including in contexts that are usually considered to measure personality). It is probably justifiable to continue referring to exploration, anxiety, and neophobia as "personality traits" and discuss plasticity with regards to personality traits, but using the same data to create measures of plasticity and measures of personality seems inherently flawed. Nevertheless, removing the "personality" measures (but preserving the use of exploration, anxiety, and neophobia tasks to measure plasticity) would be relatively simple and greatly improve the quality of the paper.

---

## Round 0.2 · accepted · Accept

I agree that it is important to consider whether behavioral flexibility is a unidimensional individual construct and whether various measures are really tapping into the same construct so I am willing to accept this version of your paper. I think the revision adequately addresses the reviewers' comments from the last round.

However, please take care of the following issues during proofs:

On line 88 please move “only” to after “exist.” Similarly on line 164.
Please use , after i.e. and e.g. (e.g., lines 174, 194, 221, 275). Check carefully for others.
On line 267, write “9” out in full, on line 261, write 5 and 3 out in full and check for other cases (e.g., line 262).
On line 335, change & to ‘and’.
The “or not” on line 387 is not needed.

#